REGISTERED REPORT PROTOCOL

# Etiology of gender incongruence and its levels of evidence: A scoping review protocol

**Juan Pablo Rojas Saffie** [ID] *, **Nicolás Eyzaguirre Bäuerle**

Department of Psychology, Finis Terrae University, Santiago, Chile

* jrojas@uft.cl

## Abstract

### Introduction

Gender Incongruence refers to the discordance between biological sex and gender identity. Although it is possible to find literature reviews about the etiology of Gender Incongruence, almost all of these correspond to non-systematic narrative reviews, so they do not make explicit the methodology used in the collection and analysis of sources, even less its levels of evidence. In order to remedy this, we will conduct a scoping review to answer the question: what are the factors associated with gender incongruence and what level of evidence is there for each factor in the scientific literature?

### Methods and analysis

We will conduct a scoping review according to the methodology specified in the JBI Manual for Evidence Synthesis (Chapter 11) and the PRISMA extension for scoping reviews (PRISMA-ScR). Four databases will be reviewed to identify papers that match our search criteria, followed by a screening of titles and abstracts, the complete reading of those articles that have not been excluded, and the coding of these using the data extraction instrument developed for this research (see S1 Appendix). Data extracted will be analyzed in terms of frequency counts of factors, types of factors and levels of evidence for each factor. Results will be presented in tabular or diagrammatic forms supported by a narrative summary.

### Findings

The present review will help to map the factors associated with incongruence between biological sex and gender identity, specifying their levels of evidence. This evidence-based knowledge will be useful for clinicians evaluating gender incongruence, especially given that international guidelines recommend careful assessment of factors that may interfere with the clarity of gender identity development and decision making.

**Data Availability Statement:** All relevant data from this study will be made available upon study completion.

**Funding:** The author(s) received no specific funding for this work.

**Competing interests:** The authors have declared that no competing interests exist.

# Introduction

## Rationale

In 1923, Magnus Hirschfeld introduced the concept of "transsexual" into professional literature to refer to a patient who presented a discordance between biological sex and gender identity [1]. He proposed that both homosexuality and transsexuality were due to natural sexual variability, explained by a set of hormonal factors originated in what he called "glandular orchestra" [2]. Since then, many researchers have tried to understand this psychological phenomenon, intending to answer what leads an individual to experience this incongruence.

The discordance between biological sex and gender identity has been conceptualized in multiple ways in the scientific literature: Transsexualism (DSM III), Transgenderism, Gender Identity Disorder (DSM IV and ICD X), Gender Dysphoria (DSM V), and Gender Incongruence (ICD XI). Although these definitions differ from each other in some nuances [3, 4], they all have in common the reference to the discordance pointed above. For this reason, in this article, we will use the concept of "Gender Incongruence" to refer to any of them.

Throughout these almost 100 years of research, significant correlations have been found between Gender Incongruence (GI) and biological, social and psychological phenomena, some of which have led researchers to propose etiological hypotheses of great interest. Some have proposed that gender incongruence stems from non-normative exposure to hormones in the womb. According to this view, the excess of testosterone in the female, or its defect in the male, could shape the brain in an unusual way, inclining these subjects toward the formation of gender incongruence [5]. While others have postulated that GI could be induced by the family system. From this perspective, gender incongruence would be a response by the child to resolve an unconscious parental conflict, allowing for system homeostasis [6]. There are also some theorists who postulate GI as a consequence of some clinical condition. For example, it has been postulated that the obsessiveness and rigidity of patients with ASD could lead them to a fixation with gender issues, facilitated by their difficulty in tolerating ambiguity [7]. Finally, there are others who propose that multiple factors must be present for a GI to develop, and that each case may be uniquely configured [8].

In an initial literature search, that is to say, prior to the development of this protocol, we found a dozen literature reviews and book chapters devoted to the etiology of GI [9–21]. Along with these reviews, other publications address the GI in a general way, devoting some sections to the subject [8, 22–31].

Although these reviews are valuable, they do have some limitations. Firstly, none were conducted systematically, which makes it difficult to know how the literature search was conducted and what inclusion criteria were followed to make its selection. Different search strategies and criteria may explain why these publications report different results. Secondly, the level of evidence for each proposed factor is not made explicit. This omission hinders the reader from adequately assessing the findings since different quality results are presented as equivalent.

Our review of the etiology of GI attempts to correct the limitations stated by the use of a systematic methodology that evidences the search strategies and inclusion criteria, among other variables. In addition, the level of evidence for each of the factors found will be specified to facilitate their appropriate consideration.

According to this, and considering our research question detailed in the next section, the main objectives of this review are (1) to account for the possible causes, factors and correlations that the scientific literature indicates in its effort to explain the genesis of GI, and (2) to detail the levels of evidence for each of the sources included in this work. This review also allows (3) to point out the knowledge gaps in the scientific literature. Finally, it is hoped that

the results of this research (4) may be useful for therapists, in line with the Standards of Care (8th edition) of the World Professional Association for Transgender Health (WPATH), where it is recommended to "identify and exclude other possible causes of apparent gender incongruence prior to the initiation of gender-affirming treatments" [32].

## Methods

The methodology for systematic reviews that best addresses the proposed objectives is the Scoping Review, executed under the guidelines proposed by the Joanna Briggs Institute (JBI) [33]. This type of review is mainly used to map the key characteristics or factors linked to a concept. In addition, they are useful to identify knowledge gaps [33]. Although, by their nature, Scoping Reviews are not designed for the critical evaluation of the evidence, their usefulness to identify the quality of the evidence [34] allow us to account for the methodological designs and the levels of evidence of the included literature.

In this context, and in accordance with the methodology of this type of systematic review, we will follow the five steps around which the Scoping Review is organized, namely: (I) identify the research question, (II) locate the appropriate academic articles, (III) select the studies that meet our inclusion criteria, (IV) chart the data from the retrieved articles, and (V) report the results.

### I. Research question

Thus, the question arising from the problem to be addressed, and the methodology best suited for it, can be expressed in these terms: What are the factors associated with gender incongruence and what level of evidence is there for each factor in the scientific literature?

### II. Locate academic articles

To identify and subsequently map the literature on Gender Incongruence etiology, this review will include a broad spectrum of scientific articles, from meta-analysis to letters to editor, as it will be detailed in the Data charting process. Gray literature will not be considered. The databases consulted to retrieve articles published in scientific journals include PubMed, Scopus, PsycINFO and CINAHL. In addition, the cited sources will be reviewed to find keywords that guide the final literature search. Articles not retrieved from these databases will be obtained by contacting the main author.

There will be no restrictions on methodology or level of evidence, except for the literature reviews, in order to avoid duplication of data. There will be no restriction on the year of publication. The language of the publications will be restricted to English and Spanish.

### III. Inclusion criteria

Papers that meet the PCC mnemonic (Population, Concept and Context) criteria stated below will be included:

**Population.** The participants of interest for this review will be those whose biological sex and gender identity are incongruent, whether they have been diagnosed by experts -with or without the use of psychometric instruments- or who self-identify as transgender. No filters for age, nationality, ethnicity, comorbidity, or biological sex will be applied.

**Concept.** This review will consider the studies that point to the etiology of the incongruence between biological sex and gender identity. Along with *etiology and aetiology*, we will also include other concepts that refer to the genesis of GI, such as *factors*, *associated factors*, *causes and influences*. Regarding GI, all concepts that point to the discordance between biological sex

and gender identity will be included: *Transsexualism*, *Transgenderism*, *Sexual Identity Disorder*, *Gender Identity Disorder*, *Gender Dysphoria* and *Gender Incongruence*. The concepts of sexual identity and gender identity will also be included to the extent that the factors involved in their development help to elucidate the origin of GI.

**Context.** In order to make the research as panoramic as possible, we have chosen to include the studies of interest regardless of the context in which they focused their activity or were carried out.

**Study selection.** As recommended in the Manual for Evidence Synthesis of the Joanna Briggs Institute [33], we will rely on a three-stage search strategy for identifying literature of interest. An initial limited search on PubMed will be undertaken followed by a screening of the text words in the title and abstracts and the index terms (see S2 Appendix). Subsequently, a second search will take place in PubMed,Scopus, PsycINFO and CINAHL using the identified terms. The third stage will consist of reviewing the reference lists of the sources that will be selected from full-text.

The identified records will be uploaded to Zotero to eliminate duplicates. Two independent researchers will review the titles and abstracts based on the PCC criteria detailed above using Rayyan (www.rayyan.ai), a web application designed to assist systematic reviews in the screening phase and to minimize the risk of bias. Articles that are approved by both reviewers will proceed to the next phase. Those in which there is disagreement will be submitted for discussion. If there is no agreement, a third person will be consulted.

Selected articles will be evaluated by full-text reading. Those that do not meet the previously stated criteria will be excluded. As in the previous case, discrepancies will be resolved by consensus or consulting a third person. To ensure transparency and reproducibility, the review finding will be reported in accordance to the Preferred Reporting Items for Systematic reviews and Meta-Analyses extension for Scoping Reviews (PRISMA-ScR) Checklist [35] (see S3 Appendix), while a flow chart of the screening process will be documented using a PRISMA diagram, as shown in Fig 1.

## IV. Data charting process

A data extraction instrument was developed based on the one suggested by the JBI (S1 Appendix). The information retrieved will include:

1. Author/s

2. Year of publication

3. Title

4. Journal

5. Volume

6. Issue

7. Pages

8. Sample's Country

9. Context

10. Methodological design

11. Summary of the contribution of the source for understanding the etiology of GI

12. Biological factors suggested in the manuscript (if any)

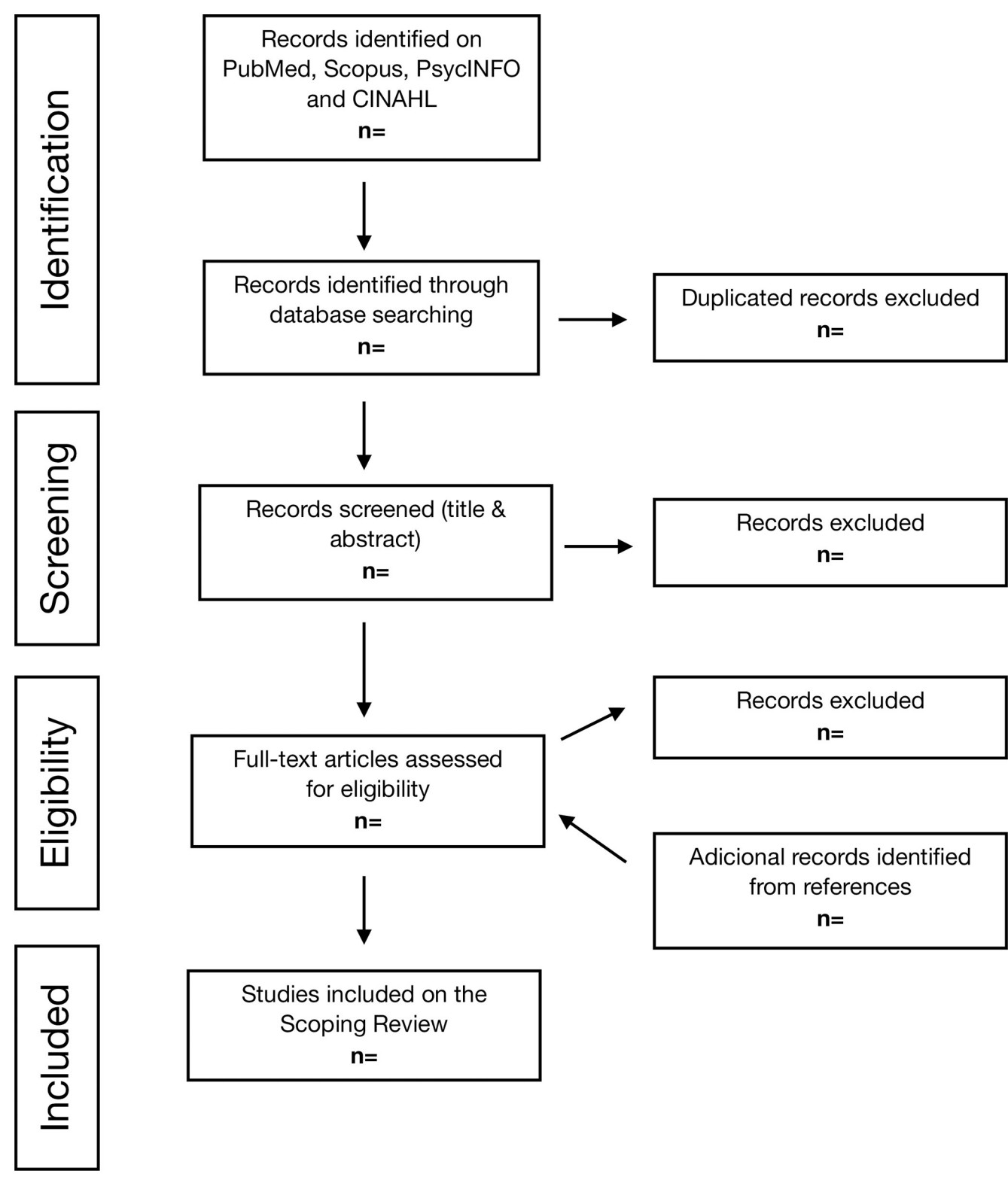

**Fig 1. PRISMA flow diagram.**

13. Social/cultural factors suggested in the manuscript (if any)

14. Psychological/internal factors suggested in the manuscript (if any)

15. Other factors suggested in the manuscript (if any)

16. Level of evidence for each factor suggested in the manuscript.

This task will be carried out independently by reviewers, with the possibility of discussing doubts that may arise during the mapping process. Any modification of the data extraction instrument will be made explicit in the scoping review.

Given that there does not seem to be unanimity in the literature on how to classify the levels of evidence, in this review we will use the following scale, ordered according to the type of methodological design: meta-analysis and systematic reviews (level 1), randomized controlled trial studies (level 2), cohort studies (level 3), case-control studies and cross-sectional studies (level 4), case series or single case report studies (level 5), literature reviews, expert opinions, editorials, letters to the editor and others (level 6) and finally, animal and laboratory studies (level 7).

## V. Results

Data extracted will be analyzed in terms of a frequency counts of factors and levels of evidence, in four ways: (1) factors associated with gender incongruence, (2) types of associated factors (biological, social/cultural and psychological/internal), (3) levels of evidence of the associated factors and (4) levels of evidence for each type of factor. These results will be presented in tabular or diagrammatic forms. A systematic narrative synthesis will be provided with information presented in the text and tables to summarize and explain the characteristics and findings of the included studies.

## Discussion

Although reviews on the etiology of gender incongruence have already been published, to our knowledge this is the first time this topic has been addressed systematically. Furthermore, this review would be the first to make explicit the level of evidence from the sources, which would allow readers to form an adequate impression of the state of the art, its strengths, weaknesses and also its gaps.

In the chapter on adolescents of the Standards of Care for the Health of Transgender and Gender Diverse People (8th edition), the following is stated: "A provider's key task is to assess the direction of the relationships that exist between any mental health challenges and the young person's self-understanding of gender care needs and then prioritize accordingly" [32]. In this sense, the study of the factors associated with gender incongruence is a contribution to clinical assessment, even more so considering that: "[. . .] mental health can also complicate the assessment of gender development and gender identity-related needs. For example, it is critical to differentiate gender incongruence from specific mental health presentations, such as obsessions and compulsions, special interests in autism, rigid thinking, broader identity problems, parent/child interaction difficulties, severe developmental anxieties [. . .], trauma, or psychotic thoughts. Mental health challenges that interfere with the clarity of identity development and gender-related decision-making should be prioritized and addressed" (Id).

Since the beginning of the last decade, treatment for gender incongruence has consisted primarily in gender-affirming medical care. However, in the last 24 months, several international health authorities have raised concerns over the uncertain risk-benefit ratio of using hormonal interventions (specifically "puberty blockers" and cross-sex hormones) as the first-line

treatment approach for young people under 18, and are restructuring their systems to prioritize psychotherapy as the first line of treatment [36–43]. Even the principal investigator of the Dutch model has stated that "an individualized approach can be offered that differentiates who will benefit from medical gender affirmation and for whom (additional) mental health support might be more appropriate" [41]. Moreover, the Dutch team itself has reached similar conclusions: "To ensure that each adolescent receives the treatment that best suits them, it is important to thoroughly explore all aspects of gender and general functioning with all adolescents before making decisions about further treatment. The conclusion of a previous study that gender-affirming treatment earlier in life may have benefits is not necessarily founded for everyone" [42].

These affirmations are consistent with the idea that gender incongruence responds to multiple etiologies, that these may be unique to each case, and that therefore, not everyone benefits from gender-affirming medical care as a first-line treatment. Understanding the etiology of GI would help clinicians decide which type of intervention will be helpful in each case. Moreover, considering that the cause of some regrets may be due to "excessive and hasty medicalization" [44], it is possible that this may also help in the prevention of detransition.

## Supporting information

**S1 Appendix. Data extraction instrument.**
(DOCX)

**S2 Appendix. Initial search (PubMed).**
(DOCX)

**S3 Appendix. Preferred Reporting Items for Systematic reviews and Meta-Analyses extension for Scoping Reviews (PRISMA-ScR) checklist.**
(DOCX)

## Acknowledgments

We would like to thank Roberto D'Angelo, Verónica Loewe and Antonia Muzard for their guidance and recommendations for the preparation of this protocol.

## Author Contributions

**Conceptualization:** Juan Pablo Rojas Saffie.

**Data curation:** Nicolás Eyzaguirre Bäuerle.

**Investigation:** Juan Pablo Rojas Saffie, Nicolás Eyzaguirre Bäuerle.

**Methodology:** Juan Pablo Rojas Saffie, Nicolás Eyzaguirre Bäuerle.

**Project administration:** Juan Pablo Rojas Saffie.

**Supervision:** Juan Pablo Rojas Saffie.

**Validation:** Juan Pablo Rojas Saffie.

**Visualization:** Juan Pablo Rojas Saffie, Nicolás Eyzaguirre Bäuerle.

**Writing – original draft:** Juan Pablo Rojas Saffie, Nicolás Eyzaguirre Bäuerle.

**Writing – review & editing:** Juan Pablo Rojas Saffie, Nicolás Eyzaguirre Bäuerle.

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
