## [Decision Letter · Decision Letter 0]

14 Dec 2022

PONE-D-22-26911Etiology of Gender Incongruence and its levels of evidence: a scoping review protocolPLOS ONE

Dear Dr. Rojas Saffie,

Thank you for

We look forward to receiving your revised manuscript.

Kind regards,

Johannes Hönekopp

Academic Editor

PLOS ONE

Journal Requirements:

4. Please upload a new copy of Figure 1 as the detail is not clear. Please follow the link for more information: " ext-link-type="uri" xlink:type="simple">https://blogs.plos.org/plos/2019/06/looking-good-tips-for-creating-your-plos-figures-graphics/"
https://blogs.plos.org/plos/2019/06/looking-good-tips-for-creating-your-plos-figures-graphics/

Reviewers' comments:

Reviewer's Responses to Questions

**Comments to the Author**

1. Does the manuscript provide a valid rationale for the proposed study, with clearly identified and justified research questions?

Reviewer #1: Yes

Reviewer #2: Yes

2. Is the protocol technically sound and planned in a manner that will lead to a meaningful outcome and allow testing the stated hypotheses?

Reviewer #1: Partly

Reviewer #2: Partly

3. Is the methodology feasible and described in sufficient detail to allow the work to be replicable?

Reviewer #1: Yes

Reviewer #2: Yes

4. Have the authors described where all data underlying the findings will be made available when the study is complete?

Reviewer #1: Yes

Reviewer #2: No

5. Is the manuscript presented in an intelligible fashion and written in standard English?

Reviewer #1: Yes

Reviewer #2: Yes

6. Review Comments to the Author

You may also provide optional suggestions and comments to authors that they might find helpful in planning their study.

Reviewer #1: This is a well-written and clearly justified protocol for a forthcoming systematic review. The research question has a good rationale which is well explained. The methods could be developed a little further, as well as the planned reporting of results. Overall I am looking forward to seeing the final review published. I have some specific points below.

Abstract

- reconsider use of term ‘narrative review’ as a systematic review can include narrative synthesis. The use of a narrative approach does not in itself obviate systematicity.

- RQ in abstract: what are the associated factors with gender incongruence and its levels of evidence in the scientific literature? It’s not clear to me what ‘its’ refers to and ‘levels of evidence’ seems to need definition. See my below comment in Methods for suggested rephrasing.

- You have made no mention of planned analysis method. I think some indication should be given here.

- Overall I don’t think the abstract goes far enough to describe methods or even make a case for how it could be useful. It is just too vaguely stated at present.

Introduction

- p8, para 3: ‘In the review prior to the development of this protocol’ – not clear what this refers to. Is this your own lit search in planning the present study? In which case, make that more clear.

- Overall very clear and well written.

Method

- p9, para 2 – couple of grammatical tweaks. Point (III) should read ‘chart’ not ‘charting’, and the last clause, ‘as will be depicted as follows’ doesn’t make sense.

- I RQ – re-phrase first sentence – ‘The question that arises from the above is the next:…’.

- I RQ – RQ itself also needs to be rephrased. Currently: What are the factors associated with gender incongruence and its levels of evidence in the scientific literature? Suggest changing to ‘What are the factors associated with gender incongruence and what level of evidence is there for each factor in the scientific literature?

- Pubmed and Scopus, although large, are not likely to generate all the relevant literature. What about PsycINFO and CINAHL? You may be interested in two of my papers published in PLOS Global Public Health, which is currently too new to be indexed. (Thompson et al ‘A PRISMA systematic review of adolescent gender dysphoria literature’ – parts 1 and 2 published, part 3 under review.) Please don’t take this to mean I expect you to cite these – this is not the point. I just want to say there may be relevant papers outwith Pubmed and Scopus so I would seriously consider casting a wider net.

- ‘In addition, the cited sources will be reviewed to capture gray literature and find keywords that guide the final literature search.’ I am not sure this would be sufficient. For grey literature, authors should consider also using databases especially designed around this literature, e.g., OpenDOAR, PsycEXTRA.

- p9 – explicate PCC in reference to mnemonic criteria

- search strategy (appendix 1) – should you include the alternative spelling ‘aetiology’ as well?

- Search strategy (appendix 1) – I appreciate this will be a preliminary search, but I suggest including terms such as ‘gender nonconforming’ ‘gender reassignment’ ‘sex reassignment’ and ‘transsexual’.

- ‘of examining the reference lists before carrying out an intensive search in both repositories’ – suggest change to ‘hand search’ and it’s not clear what is meant by an intensive search of both repositories. Does it mean your search terms may be further augmented?

- Resolving disagreement by discussion – is there a third person who could be brought in where it’s difficult to reach agreement?

- Fig 1 image quality is poor.

- p11 – levels of evidence. This hierarchy seems reasonable, but I wonder if you will be using a systematic quality assessment tool as well. That is, you could have a paper that reports an RCT (so rated 2 on level of evidence) but the quality of the study / paper is not very good. How will this be dealt with? There are good tools for assessing the quality of studies of diverse designs, such as the Crowe Critical Appraisal Tool (CCAT) or the Quality Assessment Tool for Studies of Diverse Designs (QATSDD).

Results – I appreciate the need to be vague without knowing the nature of the findings. But I think you could say that the study characteristics will be given in a table (you could even provide headings), and that you will endeavour to find a way to present the factors identified cross-referenced with their level of evidence AND an assessment of the quality of the literature (see my previous comment).

Contributions statement – I’m not sure ‘redacted’ is the term you want here. Drafted and edited?

Reviewer #2: General comments:

- Figure 1 is quite blurry at the current uploaded resolution

- etiology is almost certainly multifactorial, though I agree a scoping review could summarize potential contributors

Research question:

- Regarding this sentence, "Finally, it is hoped that the results of this research (4) may be useful for therapists...," WPATH SOC version 8 has removed the requirement that therapists are the only providers who have to evaluate and provide treatment recommendations for gender incongruence. It may be more accurate to change "therapists" to "clinicians evaluating gender incongruence"

Methods:

- With regards to the data extraction from included studies, it's unclear to me how the "methodological design" will be different from the "levels of evidence" since you are essentially substituting the method design for levels of evidence anyways.

Discussion:

- Regarding this sentence in your discussion: "One of its key points states: "There is lack of consensus and open discussion about the nature of gender dysphoria and therefore about the appropriate clinical response" [36]." I disagree that there is lack of consensus around the appropriate clinical response - the appropriate clinical response is to support trans/gender diverse individuals and there is evidence base for gender-affirming treatments in reducing mental health symptoms, improving quality of life, etc.

- With regards to this sentence in your discussion: "Understanding the etiology of gender dysphoria, would help clinicians decide which type of intervention will be helpful in each case, and might also help to prevent regret and detransition, which appears to be becoming increasingly common [37, 38]," your included references do not state that regret and/or detransition are becoming more common and they do not even speak to the incidence/prevalence. Rather those two cited studies seek to understand the phenomena of detransition. The last part of this sentence should be changed - as written, it is not supported by references and seems provocative.

- I also wonder if you could speak to how interventions might differ due to the potential etiology of gender incongruence?

7. PLOS authors have the option to publish the peer review history of their article (what does this mean?). If published, this will include your full peer review and any attached files.

Reviewer #1: **Yes: **Lucy Thompson

Reviewer #2: No

---

## [Author Response · Author response to Decision Letter 0]

13 Feb 2023

The requested information can be found in the file "Response to Reviewers".

---

## [Editor Report · Decision Letter 1]

1 Mar 2023

Etiology of Gender Incongruence and its levels of evidence: a scoping review protocol

PONE-D-22-26911R1

Dear Dr. Rojas Saffie,

We’re pleased to inform you that your manuscript has been judged scientifically suitable for publication and will be formally accepted for publication once it meets all outstanding technical requirements.

Kind regards,

Johannes Hönekopp

Academic Editor

PLOS ONE

---

## [Editor Report · Acceptance letter]

3 Mar 2023

PONE-D-22-26911R1 

Etiology of Gender Incongruence and its levels of evidence: a scoping review protocol 

Dear Dr. Rojas Saffie:

I'm pleased to inform you that your manuscript has been deemed suitable for publication in PLOS ONE. Congratulations! Your manuscript is now with our production department. 

Kind regards, 

on behalf of

Dr. Johannes Hönekopp 

Academic Editor

PLOS ONE